# Discovering the Bioactive and Antibacterial Potential of Essential Oils from Aromatic Plants of Northeastern Peru

**DOI:** 10.3390/molecules30214236

**Published:** 2025-10-30

**Authors:** Frank Fernandez-Rosillo, Elza Aguirre, Lenin Quiñones Huatangari, Segundo G. Chavez, Aline C. Caetano, Angel F. Iliquin-Chavez, Miguelina Z. Silva-Zuta, Efraín M. Castro-Alayo, César R. Balcázar-Zumaeta

**Affiliations:** 1Grupo de Modelamiento y Simulación de Procesos en la Industria Alimentarias, Instituto de Investigación de Ciencia de Datos (INSCID), Universidad Nacional de Jaén (UNJ), Carretera Jaén—San Ignacio KM 24, Cajamarca 06801, Peru; 2Programa de Doctorado en Ingeniería de Alimentos, Escuela de Posgrado, Urb. Av. Universitaria s/n, Chimbote 02712, Peru; 3Facultad de Ingeniería, Universidad Nacional del Santa (UNS), Urb. Av. Universitaria s/n, Chimbote 02712, Peru; eaguirre@uns.edu.pe; 4Facultad de Ingeniería Zootecnista, Biotecnología, Agronegocios y Ciencia de Datos, Universidad Nacional Toribio Rodríguez de Mendoza de Amazonas, Chachapoyas 01001, Peru; lenin.quinones@untrm.edu.pe; 5Instituto de Investigación para el Desarrollo Sustentable de Ceja de Selva (INDES-CES), Universidad Nacional Toribio Rodríguez de Mendoza de Amazonas, Chachapoyas 01001, Peru; segundo.quintana@untrm.edu.pe (S.G.C.); aline.caetano@untrm.edu.pe (A.C.C.); 6Instituto de Investigación, Innovación y Desarrollo para el Sector Agrario y Agroindustrial (IIDAA), Facultad de Ingeniería y Ciencias Agrarias, Universidad Nacional Toribio Rodríguez de Mendoza de Amazonas, Chachapoyas 01001, Peru; 7352940072@untrm.edu.pe (A.F.I.-C.); miguelina.silva@untrm.edu.pe (M.Z.S.-Z.); efrain.castro@untrm.edu.pe (E.M.C.-A.); cesar.balcazar@untrm.edu.pe (C.R.B.-Z.)

**Keywords:** extraction yield, antimicrobial, antioxidant, monoterpene, sesquiterpene

## Abstract

Essential oils (EOs) are mixtures of aromatic and volatile compounds. Owing to their biological properties, they are of increasing interest in the food industry as a viable alternative to natural additives. The objective of this study was to evaluate the in vitro biological activity of EOs extracted from eight plant species growing in northeastern Peru in relation to their chemical composition. EOs were extracted by hydrodistillation and evaluated for antibacterial activity, antioxidant capacity, and total phenolic content. Chemical characterization was performed by gas chromatography coupled with mass spectrometry (GC-MS), and the extraction yield was evaluated in two seasons of the year. The extraction yields varied from 0.04 to 1.50%, according to the species, with greater seasonal variation observed during the rainy season. The chemical compounds identified included monoterpene and sesquiterpene hydrocarbons, oxygenated monoterpenes and sesquiterpenes, benzene derivatives, fatty acids and derivatives, diterpenes, and phenylpropanoids. The EOs of *Magnolia jaenensis*, *Piper amalago*, *Piper glabribaccum*, and *Tesaria integrifolia* demonstrated high antibacterial activity against the Gram-positive pathogen *Staphylococcus aureus*, while the other EOs, such as *Magnolia manguillo* and *Zanthoxylum fagara*, showed intermediate activity. However, all EOs had low performance against the Gram-negative bacteria *Escherichia coli* and *Salmonella enteritidis*. The EOs from *T. integrifolia*, *Piper aduncum*, *M. manguillo*, *M. jaenensis*, and *P. glabribaccum* had high antioxidant activity. The EOs with the best biological performance were *T. integrifolia*, *M. jaenensis*, and *P. glabribaccum.*

## 1. Introduction

Peru has a great plant diversity, comprising 8% of the world’s plant diversity, that is, around 20,000 plant species, most of which are endemic [1,2]. Each region of Peru has a characteristic plant wealth, exploited for its biological properties in traditional medicine, aromatherapy, preparation, and preservation of local foods in rural and urban areas [3,4]. Jaen is one of the 13 provinces that make up the department of Cajamarca. It is located in the north of the department and has an area of 5232 km^2^. The extension of this province is made up of diverse ecosystems, with dry and humid tropical forests that harbor native and non-native plant species [5,6].

The Asteraceae family is distributed across almost all continents except Antarctica [7,8]. With 32,913 species distributed in 1911 genera, it is considered the richest and most biologically diverse family of angiosperms [9], which includes the species *T. integrifolia* [10,11]. On the other hand, Piperaceae is an angiosperm plant family mainly distributed in tropical and subtropical areas of America, Asia, and Africa [12]. This plant family comprises five genera, with *Piper* sp. being the most important for its medicinal and aromatic uses [13], which includes the species *P. aduncum*, *P. amalogo*, and *P. glabribaccum* [14].

On the other hand, Rutaceae is a family of flowering plants, known as the citrus family, consisting of herbs, shrubs, and small trees. The family contains 160 genera and 1900 species [15]. Of these species, *Esembeckia cornuta* and *Z. fagara* occur in seasonally dry and naturally distributed tropical forests in the Marañon River valley, northeastern Peru, the former being an endemic species [16]. In addition, Magnoliaceae is a family of angiosperms, consisting of 17 genera and about 300 species [17,18,19]. Approximately 219 species of the genus *Magnolia* are woody plants with primitive flowers. They are distributed in tropical and subtropical regions [20]. However, in this genus two new species have been described and reported in the department of Cajamarca, Peru, *M. jaenensis* and *M. manguillo*, constituting the first records in Peruvian montane forests at altitudes exceeding 2100 m in altitude [21].

The food industry is immersed in a dichotomy of production and formulation constructs, where two trends can be distinguished: the first one implies the use of synthetic additives in intensive food production systems, given their low cost and immediate availability on the market [22,23]. The second trend is more recent and depends on the use of green technologies in food production systems to yield more natural products [23]. The latter includes the application of emerging technologies or the use of ingredients and/or additives of natural origin, such as preservatives, flavors, and aroma enhancers [24]. Within the category of natural additives, EOs are gaining interest as viable alternatives because they possess various biological properties.

An EO is a complex mixture of volatile compounds formed in the secondary metabolism of plants and responsible for their characteristic aroma. They can be obtained from different parts of plants, such as leaves, stems, seeds, fruits, flowers, roots, and bark, using different extraction methods such as hydrodistillation, steam distillation, cold pressing, and supercritical fluid extraction [25,26]. The results of various studies have reported that EOs exhibit significant antibacterial effects against foodborne pathogens such as *E. coli* and *S. aureus* [25,27,28,29,30,31]. For example, the EO of the *Piper* species showed potential use as an antimicrobial and antioxidant agent [32]. In addition, EOs have strong antioxidant capacity, which helps to extend the shelf life of various food products by inhibiting oxidative degradation [33].

This study evaluated the seasonal variation in the extraction yield of EOs extracted from eight native plant species with ethnopharmacological importance in the province of Jaen (Cajamarca, Peru) during two seasons of the year, from July to September 2023 (dry season) and from February to March 2024 (rainy season). The EOs extracted during the dry season were chemically characterized by GC-MS, and their biological activities were evaluated in vitro. Antibacterial activity against three species of foodborne pathogens was evaluated using the disc diffusion assay, antioxidant capacity was evaluated using the ABTS, FRAP, and DPPH assays, and the total phenolic compound content was determined using the Folin–Ciocalteu method. The essential oils with the best biological performance were identified using the non-hierarchical k-means clustering method and principal component analysis (PCA).

## 2. Results and Discussion

### 2.1. Extraction Yield of Essential Oils

Figure 1 shows the comparison of the extraction yields with respect to the seasonal variation in the EOs; the yield data represent the average of three independent extractions (n = 3) carried out for each plant species and in each season (dry and rainy). It is evident that during the rainy season, higher yields were obtained for *E. cornuta*, *M. jaenensis*, *P. aduncum*, *P. amalago*, *P. glabribaccum*, and *T. integrifolia*, while for *M. manguillo* and *Z. fagara*, a lower yield was observed. The EO yield did not differ according to the season of sample collection. Other research has shown the yields of EOs extracted from fresh leaves by hydrodistillation for the plants studied. Setzer et al. [34] obtained an extraction yield of 0.02% for *Z. fagara* from Costa Rica, which was lower than that obtained in the present investigation (1.5%). Studies in Brazil have shown an extraction yield of 0.75% for the EO of *P. amalogo* [35], a value lower than that shown in the present study (0.83%). Bergo et al. [36] determined the extraction yield of *P. aduncum* to be lower than 0.29%, which differs from the results of the present study (0.66%). This shows that plants growing in the province of Jaen could be an important source of EOs for the industry.

Regarding the seasonal variation in extraction yield, the results are in agreement with Ahrar [37], who reported a variation in the yields of *Mentha longifolia* L. EO. Smitha & Tripathy [38], who demonstrated that the content and chemical composition of EOs obtained from different *Ocimum* species varied widely according to geographical location, harvest time, and growth stage. Perigo et al. [12] showed that the content and composition of oils extracted from 11 *Piper* species were influenced by plant species and environmental conditions. However, the results obtained here concerning the higher production and yield of EOs for most plant species is in agreement with the data reported by Liao et al. [39], who indicated that the yields of EOs extracted from *Lavandin* in different seasons exhibit lower yields in winter and the highest in summer, and with Pinheiro et al. [40], who evaluated the effect of seasonality on the yield and chemical composition of *Hesperozygis ringens* (Benth.) Epling EO; the highest yields were obtained in autumn, spring, and summer, while the lowest yields were obtained in winter. However, the elicitation of secondary metabolites and the composition of complex mixtures such as EOs are also influenced by environmental factors such as pollution, climate, and diseases [41]. The variation in extraction yield exhibited is due to the different edaphoclimatic conditions of their habitats (dry and humid forest); the processes of biosynthesis of compounds present in EOs in the metabolism of plants are influenced by climate, rainfall, soil, altitude, exposure to sunlight, and the developmental stage of the plant [42,43,44,45], determining the storage at different concentrations of EOs in glandular trichomes [46].

### 2.2. Volatile Profile of Essential Oils

Table 1 reveals a remarkable richness of sesquiterpenoids, particularly bicyclic types like caryophyllene and its derivative, which are present in all the species evaluated. Key compounds such as τ-cadinol and the occurring sesquiterpene alcohols (e.g., the isomer of guaiol and eudesmol) showed a widespread presence. Phenylpropanoids like apiol are also significant, especially in the *Piper* species and *Magnolia jaenensis*. This organized data is crucial for identifying chemotaxonomic patterns and potential biological activities linked to specific compound classes across these native Peruvian plants.

The properties of the volatile compounds of EOs, such as hydrophobicity and reactivity, cause the breakdown of the lipid structures of biological membranes, which alters and impairs cellular functions, making possible a potential technological application as antibacterial agents [47]. There are several reports on the chemical composition of the EO of *P. aduncum* extracted from leaves by hydrodistillation. Cossolin et al. [48] reported myristicin (30.03%), aromadendrene (9.20%), dillapiol (8.43%), α-serinene (7.31%), tridecaene (6.26%), γ-elemene (4.58%), and *o*-cymene (4.20%) as the major compounds. Silva et al. [49] reported dillapiol (52.37%) and γ-terpinene (8.98%), whereas Oliveira et al. [50] reported 1,8-cineole (53.9%), α-pinene (12.7%), β-pinene (8.5%), and *trans*-ocimene (5.7%). Santana et al. [51] reported (*E*)-isocroweacin (29.52%), apiol (28.62%), and elemicin (7.82%). Jaramillo-Colorado et al. [52] obtained dillapiol (48.2%) and 1,8-cineol (11.4%), and Mamood et al. [53] reported apiol (38.01%), methyl isobutyl ketone (8.26%), piperitone (3.34%), and caryophyllene (2.45%).

Some reported compounds coincide with those found in the present investigation, such as myristicin (38.26%), caryophyllene (11.10%), isoaromadendrene epoxide (8.40%), pentadecane (5.86%), β-copaene (3.12%), and apiol (3.08%). Furthermore, our results are in agreement with the literature, in which dillapiol, myristicin, and caryophyllene are the main components of *P. aduncum* EO [54,55,56]. Studies have been reported in Brazil on the chemical composition of the EO of *P. amalogo*; for example, da Silva Mota [57] showed chemical composition analysis by GC-MS for the EO extracted from leaves by hydrodistillation, in which they identified 52 components, with the main components being α-amorphene (25.7%), *p*-cymene (9.4%), and (*E*)-methylgeranate (7.8%). Potzernheim et al. [58] showed α-pinene (30.5%), camphene (8.9%), and limonene (6.8%), and [59] reported EC elemene (36.5%), caryophyllene oxide (18.0%), caryophyllene E (17.8%), bicyclogermacrene (16.4%), germacrene D (10.9%), and α-pinene (9.30%) as major components; similarly Morandim-Giannetti et al. [60] showed the predominance of γ-muurolene (7.27%), germacrene D (9.94%), bicyclogermacrene (27.91%), spatulenol (19.22%), and α-cadinol (7.6%); Ferraz et al. [61] also reported limonene (20.52%), δ-elemene (6.82%), and zingiberene (11.18%) as main components, which present some coincidences with the present work where γ-muurolene, β-pinene, and caryophyllene are the matching compounds.

For the EO of *T. integrifolia* there are scientific reports that show some constituents isolated from the leaf extract. For example, Feo et al. [62] reported 3,4-dicaffeoylquinic acid (0.020%), 3,4,5-tricaphoylquinic acid (0.013%), quercetin (0.017%), quercetin-3-*O*-glucoside (0.012%), rutin (0.020%), and naringin (0.025%), derived from caffeoylquinic acid [63]; Silva-Correa et al. [64] and Ono et al. [65] obtained sesquiterpene compounds of the eudesmane type, compounds that differ from what was found in this study.

In addition, among the studies of the chemical composition of the EO of *Z. fagara*, the analysis of Setzer et al. [34] identified 23 compounds of the EO from Costa Rica, with the most abundant components being citronellol (26.1%), geraniol (15.3%), citronellal (11.3%), geranial (11.6%), and neral (9.6%). Likewise, Pérez-López et al. [66] reported that for the EO extracted in Mexico, the main compounds were silvestrene (25.3%) and E-caryophyllene (23.6%), constituting the main components of sesquiterpene and monoterpene hydrocarbons with 51.1 and 37.5%, respectively. Pino et al. [67] reported the chemical composition of EO in Cuba, where they identified 37 compounds, of which α-bisabolol (11.3%) and bulnesol (8.7%) were the most abundant. Prieto et al. [68] determined the chemical composition of EOs isolated from fruits in Colombia by steam distillation, identifying 57 compounds, and the main constituents contained germacrene D-4-ol (21.1%), elemol (8.35%), and α-cadinol (8.22%), which are different from those found in this study. Consequently, within the identified compounds we have only two matches with different percentages of relative abundance between caryophyllene with 0.50 and 40.00% and α-cadinol with 0.70 and 1.00% for Setzer et al. [34] and the present investigation, respectively.

### 2.3. Evaluation of the Antibacterial Activity of Essential Oils

Table 2 shows the results of the disk diffusion method, where the averages of the inhibition halos for each bacteria corresponding to the treatment with the EOs, the control (streptomycin), and the PIR are shown. The results revealed different levels of inhibition. The EOs of *M. jaenensis*, *P. amalago*, *P. glabribaccum*, and *T. integrifolia* exhibited high antibacterial activity against the major positive bacteria *S. aureus*; however *M. manguillo* and *Z. fagara* showed intermediate activity, whereas against the major negative bacteria *E. coli* and *S. enteritidis* all EOs showed low activity.

Our results were similar to those reported by Perigo et al. [12], in which EOs from most of the 11 *Piper* species investigated showed inhibitory activity against pathogenic bacteria in vitro; limonene and *cis*-β-ocimene were associated with the inhibition of *S. aureus*, thus demonstrating the chemical diversity of *Piper* EOs and their potential as new antibacterial agents in various industrial applications. However, the present investigation differs from the results reported by Braga Carneiro et al. [69], because in their study the EO extracted from *P. aduncum* leaves revealed potential antibacterial activity associated with the presence of its main compound, dillapiol, contrary to what was reported in the research on the EO of this species, which showed low antibacterial activity against the bacteria evaluated. For the EO of *P. amalogo*, [70] revealed that the methanolic extract of this species exhibited weak activity against *E. coli* and *S. aureus*. Araujo Baptista et al. [71] showed that the EO exhibited a moderate effect against *S. aureus* and [72] reported weak activity against *E. coli* and *S. aureus*; these results differ from those exhibited in the present investigation, where the EO of this species showed high bactericidal activity against *S. aureus*. However, in this study, EOs were evaluated in their pure (undiluted) form as an initial, qualitative approach to determining the intrinsic antibacterial potential of each extract. This strategy is common in exploratory screening stages, as it allows for the rapid and effective identification of those essences with the most promising activity, avoiding the possible modulatory effect of a solvent that could alter the diffusion or bioactivity of the volatile components [73,74]. Even though this approach does not allow for the calculation of minimum inhibitory concentrations (MIC) and may hinder direct comparison with studies using solvents, it was specifically selected to prioritize the detection of activity in the context of an initial phytochemical and biological analysis. The results obtained should therefore be interpreted as an initial assessment of the potential of these EOs, laying the groundwork for future research that includes serial dilutions for a more accurate quantitative characterization of their antibacterial potency.

### 2.4. Evaluation of Antioxidant Capacity and Total Content of Phenolic Compounds of Essential Oils

Table 3 shows the antioxidant capacity determined by DPPH, FRAP, and ABTS assays, as well as the total content of the phenolic compounds of the EOs. A wide range of antioxidant capacity was observed, with *P. aduncum* presenting the highest value in the three antioxidant capacity tests. The species whose EO presented the highest total phenolic compound content was *T. integrifolia* with an average of 159.34 mg GAE/g sample. The study exhibited significant differences between the results of the DPPH, FRAP, ABTS, and TPC assays for each EO, showing a dependence on the chemical composition, quality, and purity of each EO [75]. The results of the total content of phenolic compounds reported in the research for the EOs ranged from 50.11 to 159.34 mg GAE/g; these results were higher than those obtained by Proestos et al. [76] for different EOs, which ranged from 1.0 to 18 mg GAE/g, as well as those reported by Lin et al. [77], in which EOs ranged from 4.05 to 57.69 (µg GAE/5 mg). Similarly, the result obtained in the ABTS assay for the EO of *P. aduncum* exhibited higher values than those reported by Guerrini et al. [13]. The sample size for each analysis was three independent replicates (n = 3) for each EO and each assay (DPPH, FRAP, ABTS, and TPC).

### 2.5. Grouping of the Best Performing Essential Oils in Terms of Antibacterial Activity and Antioxidant Activity

To explore the integrated relationships between extraction yield, antioxidant activity, and antibacterial activity, a principal component analysis (PCA) was performed using a unified data matrix containing all these variables (Table 4). The resulting biplot (Figure 2), representing the first two principal components, together explained 84% of the total variance in the data (PC1 + PC2). This percentage of cumulative variance indicates that the graph effectively captures most of the information and variability structure present in the original dataset.

The interpretation of the biplot reveals that Component 1 (horizontal axis) was positively associated with high antioxidant activity (ABTS and FRAP variables) and negatively associated with extraction yield. Component 2 (vertical axis) correlated mainly with antibacterial activity. This visual representation allowed three distinct groups of essential oils to be identified: Group 1 (*T. integrifolia*, *M. jaenensis*, and *P. glabribaccum*) was characterized by its high antioxidant and antibacterial activity; Group 2 (*P. amalago* and *Z. fagara*) grouped the species with the highest extraction yield; and Group 3 (*P. aduncum*, *E. cornuta*, and *M. manguillo*) contained the EOs with the lowest antibacterial activity.

### 2.6. Chemical Composition of Essential Oils

To explore the relationship between plant species and their dominant chemical profile, a simple correspondence analysis (CA) was performed using only the major compounds (>5%). Figure 3 represents major compounds of each EO. The graph shows that the EOs of *T. integrifolia* do not share major compounds with the EOs of the other species. Similarly, the EOs of *P. amalogo* and *P. aduncum* have no major compounds in common with other EOs. It can also be seen that two of the three species with the highest antioxidant and antimicrobial activity (*M. jaenensis* and *P. glabribaccum*) have copaene (c24) and *trans*-nerolidol (c37) as major compounds.

Due to the use of native and endemic plants from the province of Jaen (Cajamarca, Peru), reports on extraction yields, biological activity (antibacterial and antioxidant) and chemical composition of EOs extracted from these plant species are scarce in the scientific literature; e.g., Vásquez-Ocmín et al. [78] reported that *P. glabribaccum*, from the department of Loreto (Peru), is widely used traditionally as an anti-infective drug, but they do not report extraction yield, biological activity, or chemical characterization. Likewise, for *M. jaenensis*, *M. manguillo*, and *E. cornuta* there are no studies on biological activity or characterization of chemical composition, only ecological reports as threatened and endangered species are evidenced [16,21,79]. Also, *T. integrifolia* has been reported as a source of extracts other than EOs, with no reported extraction yield or biological activity.

## 3. Materials and Methods

### 3.1. Georeferencing and Collection of Biological Samples

The eight plant species were selected based primarily on their ethnopharmacological importance, specifically their traditional use in folk medicine of northeastern Peru for treating infections and inflammatory processes. This main criterion was complemented by local availability and the prioritization of native aromatic species. This approach aligned the study’s objective of scientifically validating traditional uses with the principle of revaluing the region’s ethnomedical resources, while also providing a guided framework for exploring their antibacterial and antioxidant activity.

Eight plant species were collected in situ in the districts of Jaen and Bellavista in the province of Jaen (Cajamarca, Peru). The samples were fresh leaves, which were cut from the beginning of the leaf, avoiding damage to the plant and allowing new regrowth. Leaves in poor condition were eliminated. The reference coordinates were recorded using a global positioning system with Garmin GPS, GPSMAP 64sx (Figure 4).

Harvesting was carried out in two seasons: July to September 2023 (drought season) and February to March 2024 (rainy season) (Figure 5).

### 3.2. Essential Oil Extraction

Cleaned and selected fresh leaves of each species were dried under shade for 72 h at room temperature, cut into small pieces, and crushed to obtain a homogeneous sample. Hydrodistillation was performed for 2 h using a Clevenger apparatus. The weight/volume ratio was 1 g sample/11 mL distilled water. At the end of the extraction process, the EOs were stored in amber flasks under refrigeration at 4 °C until use. The extraction yield was expressed as the ratio of the volume of EO extracted to the weight of the dry sample, multiplied by 100 [80].

### 3.3. Chemical Characterization of Essential Oils

The chemical composition of the EOs was determined by gas chromatography using an Agilent GC System Chromatograph model 7890B (Santa Clara, CA, USA), coupled to an MSD 5977 B quadrupole mass detector. The EOs was diluted (1 μL EO and 99 μL hexane) beforehand. Chromatographic separation was performed on a DB-5MS UI capillary column (60 m × 0.25 mm × 1.0 μm, Agilent J&W Scientific, Folsom, CA, USA), and helium was used as the carrier gas at a flow rate of 1 mL·min^−1^. The injection was performed in split mode (50:1), with an injection volume of 0.5 μL. The injector, detector, transfer line, and ionization source were maintained at 220 °C, 150 °C, 240 °C, and 280 °C, respectively. The oven was initially programmed to operate at 60 °C, and the temperature was increased by 3 °C·min^−1^ to 246 °C, held for 8 min, and then reached 300 °C at a rate of 5 °C·min^−1^. The mass spectra of each compound were obtained in scan mode with a mass range (*m*/*z*) of 40–600 amu. The detected compounds were identified by comparison with the National Institute of Standards and Technology library database (NIST Library 17), and their identification was confirmed by determining the retention index through the injection of the n-alkane standard (C_8_–C_20_) [81].

### 3.4. Antibacterial Activity

The antibacterial activity was performed by growth inhibition in disk diffusion assays against three bacterial species associated with foodborne diseases. The positive control for growth inhibition was streptomycin.

The bacterial strains evaluated were standard cultures obtained from the American Type Culture Collection (ATCC, Manassas, VA, USA). The bacteria evaluated were *Escherichia coli* ATCC 25922, *Salmonella enteritidis* ATCC 13076, and *Staphylococcus aureus* ATCC 25923. For the disk diffusion method, the Clinical and Laboratory Standards Institute protocol M02-A11 was used [82]. The EOs were evaluated to be pure (100% purity). Bacterial strains were grown on Tryptic soy agar (TSA) and isolated colonies of each bacterium were transferred to tubes of sterile saline (NaCl 0.85%) until an optical density of 0.08 to 0.1 absorbance at the wavelength of 625 nm, corresponding to the 0.5 McFarland standard equivalent to 1–2 × 10^8^ CFU/mL, was reached [82]. This was either the inoculum or working bacterial solution. Next, Mueller Hinton (M173, HiMedia Laboratories Pvt. Ltd, Mumbai, India) agar plates were inoculated with the respective bacterial inoculum with the aid of a sterile isopo, and 10 µL of each EO (100%) was placed on sterile paper discs (Whatman No. 1, Whatman^®^, Maidstone, UK) of 6 mm diameter, which were transferred to MH agar plates. Three disks containing the same EOs and one disk containing streptomycin antibiotic as a positive control were placed on each plate. MH agar plates were then incubated at 37 °C for 20 h. After the incubation period, the diameter of the zone of inhibition (DZI) was measured using Vernier (150 mm, 0.02 mm precision, CLA006, Uyustools, Hangzhou, China). The experiments were performed in triplicates [83]. With the values of the diameters of the halos of inhibition of the growth of the microorganisms, the percentage of the relative inhibitory effect (PIR) with respect to the positive control was calculated for each EO and microorganism; this was expressed as a percentage in relation to the mean of the diameter of the inhibitory halo versus the diameter of the inhibitory halo of the positive control [84]. Antibacterial activity was considered high when the PIR was greater than 70%, intermediate between 50 and 70%, and low when it was less than 50% [85].

### 3.5. Antioxidant Capacity

DPPH assay: For the determination of the antioxidant capacity by the DPPH assay, the methodology proposed by Scherer & Godoy [86] was used. The reagent system that was mixed contained 3 mL of ethanol, 300 µL of 150 µM DPPH radical solution in ethanol, and 500 µL of the diluted AE solutions; incubation was carried out for 45 min at 25 °C in the dark. The optical density was then measured at 517 nm (GENESYS^TM^ 150, Thermo Scientific, Waltham, MA, USA). The blank used was ethanol, and a calibration curve was constructed using the Trolox standard at concentrations between 10 and 100 µM. The assay was performed in triplicate, and the results were expressed in µmol Trolox equivalent (TE) per gram of sample (µmol TE/g) [87].

ABTS assay: To determine the antioxidant capacity by the ABTS test, the methodology proposed by Re et al. [88] was followed with some modifications. With the reaction of 88 µL of 140 mM potassium persulphate (final concentration 2.45 mM) and 5 mL of an aqueous ABTS solution (7 mM), the radical cation ABTS^•+^ was generated. Before use, the mixture was kept for 16 h in the dark and then diluted with 50% ethanol. Using a spectrophotometer, an absorbance at 734 nm of 1.0 ± 0.02 units was obtained. Then, 30 μL of each EO or Trolox (reference substance) was reacted with 3 mL of blue-green ABTS radical solution. The absorbance decay at 734 nm was measured every 6 min (GENESYS^TM^ 150, Thermo Scientific, Waltham, MA, USA). Calibration was performed using ethanolic solutions of known Trolox concentrations. The assay was performed in triplicate, and the results were expressed in µmol of Trolox equivalents (TE) per gram of sample (μmol TE/g) [89].

FRAP assay: The methodology described in Ayed et al. [90] was used as a reference, with some modifications. FRAP reagent was prepared by making a mixture of 0.3 M acetate buffer (pH 3.6), 0.1 M TPTZ diluted in 0.4 M hydrochloric acid, and 0.2 M ferric chloride hexahydrate (FeCl_3_·6H_2_O) at a 10:1:1 ratio. Then, 2.7 mL of the FRAP reagent was placed in test tubes, 90 µL of EO and 270 µL of distilled water were added, and then placed in a water bath at 30 °C for 4 min, and the absorbance was recorded at 593 nm using a spectrophotometer (EMC-11-UV, EMCLAB Instruments GmbH, Duisburg, Germany). Antioxidant capacity was determined from the linear calibration curve of ferrous sulfate heptahydrate (FeSO_4_·7H_2_O) from 200 to 3800 µm. The assay was performed in triplicate, and the results are expressed as µmol Fe^2+^/g.

### 3.6. Total Content of Phenolic Compounds

It was determined using the Folin–Ciocalteu method described by Singleton et al. [91] with some modifications. The EOs were diluted in ethanol, and the solution was homogenized for 45 min in an ultrasonic bath at 45 °C. The mixture of 150 µL of the dilution of the ECs, 750 µL of Folin–Ciocalteu reagent (1:10), and 600 µL of 7.5% Na_2_CO_3_ was incubated in the dark for 2 h at 25 °C, and then the optical density of the mixture was measured using a UV-vis spectrophotometer (GENESYS^TM^ 150, Thermo Scientific, Waltham, MA, USA) at 740 nm. A blank was prepared under conditions similar to those used for distilled water. Gallic acid as a standard was used to construct a calibration curve. The assay was performed in triplicate, and the results were shown as mg gallic acid equivalents per gram of sample (mg GAE/g) [87].

### 3.7. Data Analysis

For the evaluation of the antioxidant capacity by DPPH, FRAP, and ABTS assays, as well as the total content of phenolic compounds in the EOs, an analysis of variance and Tukey’s test were performed. Principal component analysis (PCA) was performed on the relative inhibition percentages (RIAs) to obtain an indicator (first component) that explains, as a whole, the antibacterial activity of the EOs. To explain the antioxidant activity and phenolic content, the results of the three methods (DPPH, FRAP, and ABTS) and the total content of phenolic compounds in the EOs were analyzed by PCA. The normality of the residuals was confirmed using the Shapiro–Wilk test, and homoscedasticity (homogeneity of variances) was checked using Levene’s test. Similarly, the extraction yield percentages in the two harvesting stages were subjected to PCA. With the indicators in Table 4, cluster analysis was performed using the non-hierarchical k-means clustering method and principal component analysis to identify the variables that explain the formation of each cluster. A correspondence analysis of the majority of compounds was performed to characterize the chemical composition of the EOs.

## 4. Conclusions

It was found that in the rainy season, corresponding to the collection in February and March 2024, a higher extraction yield was obtained for most of the EOs. All the oils evaluated showed low activity against *E. coli* and *S. enteritidis*, while against *S. aureus* the oils were divided into three groups exhibiting low, medium, and high activity, where the EOs of *P. amalogo*, *P. glabribaccum*, *T. integrifolia*, and *M. jaenensis* showed high activity. For antioxidant activity, the EOs of *T. integrifolia*, *P. aduncum*, *M. Manguillo*, *M. Jaenensis*, and *P. glabribacum* species showed the highest values in all tests. The chemical composition of each oil showed a distinct presence of monoterpene hydrocarbons, oxygenated monoterpenes, sesquiterpene hydrocarbons, oxygenated sesquiterpenes, and alcohols. The EOs with the best biological performance by principal component analysis in relation to their antibacterial activity, antioxidant capacity, and phenolic compound content were *T. integrifolia*, *M. jaenensis*, and *P. glabribacum*.

## Figures and Tables

**Figure 1 molecules-30-04236-f001:**
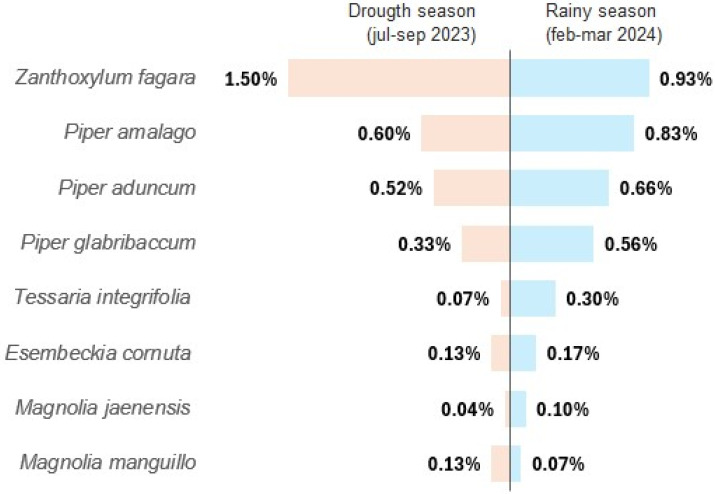
Seasonal comparison of extraction yields of essential oils.

**Figure 2 molecules-30-04236-f002:**
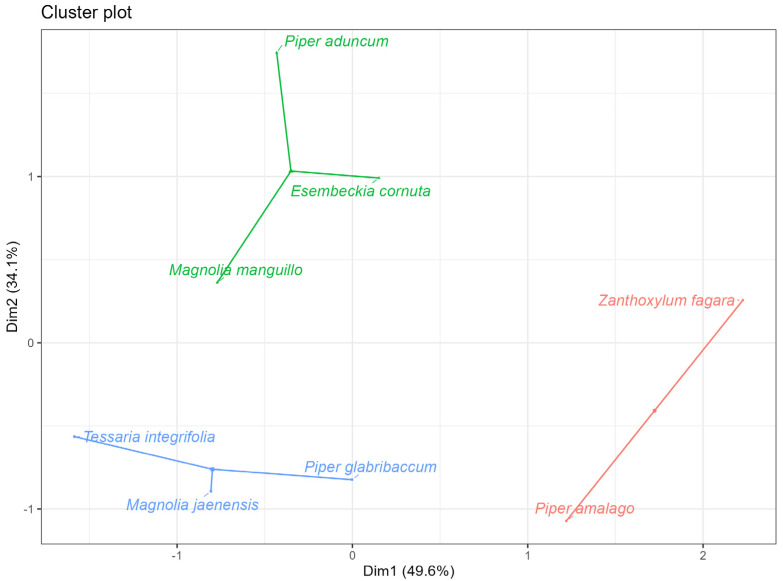
Principal component analysis (PCA) and K-means clustering of essential oils from eight plant species. Note: PCA biplot showing the distribution of EOs and variables in the space defined by the first two principal components (PC1 and PC2), which together explain 84% of the total variance in the data. The variables analyzed were extraction yield, antioxidant activity (measured by the DPPH, FRAP, and ABTS methods), total antioxidant capacity (TPC), and antimicrobial activity. The grouping resulting from the K-means analysis identifies three distinct clusters: blue represents Group 1, red Group 2, and green Group 3. The data used are reported in Table 4.

**Figure 3 molecules-30-04236-f003:**
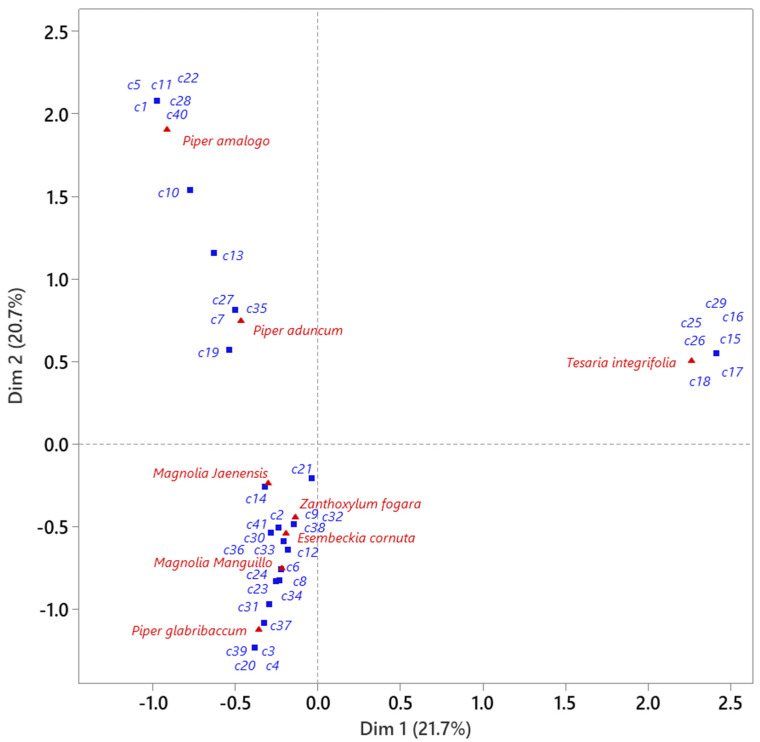
Correspondence between EOs and major compounds (>5%) using simple correspondence analysis (CA). Majority compound labels (>5%) c1: 1,4,7,-cyclondecatriene, 1,5,9,9-tetramethyl-, Z,Z,Z-; c2: (−)-*cis*-β-Elemene; c3: (−)-α-cubebene; c4: (−)-α-gurjunene; c5: (+)-δ-cadinene; c6: (1a*R*,4a*R*,7*S*,7a*R*,7b*R*)-1,1,7-Trimethyl-4-methylenedecahydro-1*H*-cyclopropa[e]azulen-7-ol; c7: (3*R*,3a*R*,3b*R*,4*S*,7*R*,7a*R*)-4-Isopropyl-3,7-dimethyloctahydro-1*H*-cyclopenta[1,3]cyclopropa[1,2]benzen-3-ol; c8: α-Cadinol; c9: α-Muurolene; c10: β-Copaene; c11: γ-Muurolene; c12: 1,5,9,9-tetramethyl-1,4,7-cycloundecatriene; c13: (1a*R*,4a*R*,7*S*,7a*R*,7b*R*)-1,1,7-trimethyl-4-methylenedecahydro-1*H*-cyclopropa[e]azulen-7-ol; c14: 2-Methyl-1-pentene; c15: 2-naphthalenomethanol, decahydro-α,α,4a-trimethyl-8-methylene-, [2*R*-(2α,4aα,8aβ)]-; c16: 4,4a,5,6,7,8-Hexahydro-4a,8-dimethylnaphthalen-2(3*H*)-one; c17: 6,7-dimethyl-1,2,3,5,8,8a-hexahydronaphthalene; c18: 3,8-dimethyl-5-α-hydroxy-δ^9-octa hydroazulene acetate; c19: Apiol; c20: Bicyclo[5.2.0]nonane, 2-methylen-4,8,8-trimethyl-4-vinyl-; c21: Caryophyllene; c22: Cyclophenchene; c23: *cis*-β-Copaene; c24: Copaene; c25: Dehydrofukinone; c26: Dihydroagarofuran; c27: Isoaromadrene epoxide; c28: Eucalyptol; c29: Phenol, 2,4-bis(1,1-dimethylethyl)-6-methyl-; c30: Phytol; c31: Guaiol; c32: Hedicariol; c33: Humulan-1,6-dien-3-ol; c34: Isospatulenol; c35: Pentadecane; c36: Peruviol; c37: *trans*-nerolidol; c38: α-epi-7-epi-5-Eudesmol; c39; α-selinene; c40: β-pinene; c41: δ-amorphene.

**Figure 4 molecules-30-04236-f004:**
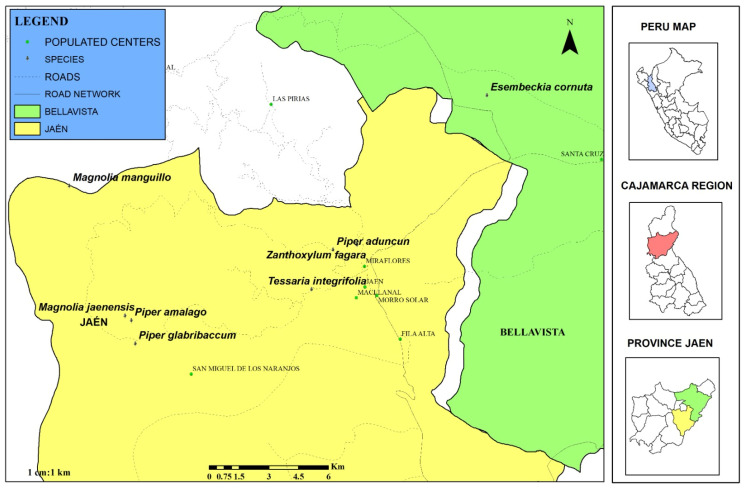
Geographical distribution of the collection points of plant material.

**Figure 5 molecules-30-04236-f005:**
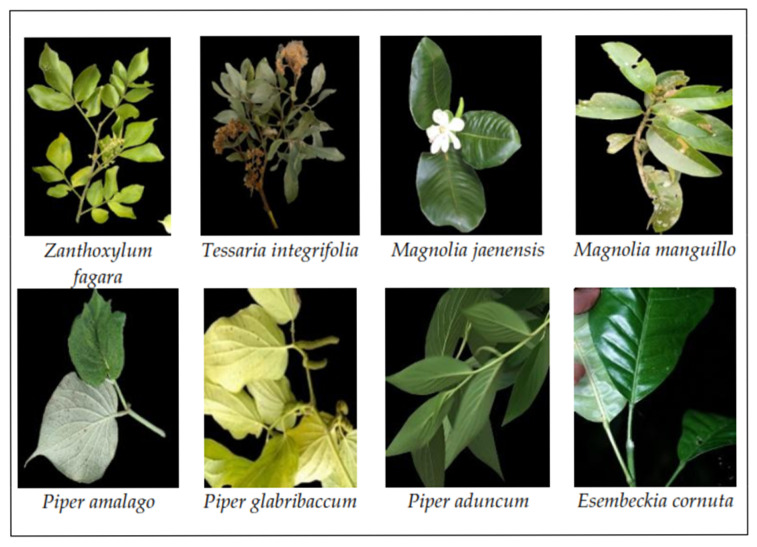
Plant species from which EOs were extracted.

**Table 1 molecules-30-04236-t001:** Compounds detected in the extracted essential oils by GC-MS.

Group	Subclass	Compound Name	Relative Abundance (%)							
			*Zanthoxylum fagara*	*Piper amalago*	*Piper aduncum* L.	*Piper glabribaccum*	*Esembeckia cornuta*	*Magnolia manguillo*	*Magnolia jaenensis*	*Tessaria integrifolia*
Monoterpenoids	Acyclic Monoterpene	β-myrcene	-	0.29	-	-	0.19	-	-	-
		citral	-	0.06	-	0.09	0.12	0.09	0.46	-
		neral	-	0.04	-	-	0.07	0.05	0.28	-
		geranial	-	-	-	-	-	-	-	-
		*trans*-2-decenal	-	-	-	-	-	-	0.02	-
	Monocyclic Monoterpene	D-limonene	0.08	0.71	-	-	-	0.25	-	-
		α-terpinene	-	0.08	-	-	-	-	-	-
		γ-terpinene	-	0.15	-	-	-	-	-	-
		α-terpinolene	-	0.08	-	-	-	-	-	-
		*p*-cymene	0.05	0.28	-	-	0.06	0.16	-	-
		α-phellandrene	0.07	-	-	-	0.07	-	-	-
		β-pinene	-	9.96	-	-	-	-	0.25	-
		sabinene	0.07	0.15	-	-	-	-	-	-
		camphene	-	0.08	-	-	-	-	-	-
	Bicyclic Monoterpene	eucalyptol (1,8-cineole)	-	4.5	-	-	0.11	-	0.14	-
		L-α-terpineol	-	1.73	-	-	0.17	-	0.11	-
		(−)-terpinen-4-ol	-	0.4	-	-	-	-	-	-
		α-terpineol	-	-	-	-	-	0.07	-	-
		4-terpineol	-	-	-	-	-	0.54	-	-
		(−)-borneol	-	0.16	-	-	-		-	-
		(−)-myrtenol	-	0.1	-	-	-	0.11	-	0.09
		myrtenal	-	-	-	-	-	0.19	-	-
		pinocarvone	-	-	-	-	-	-	-	0.11
		isobornyl acetate	-	-	-	-	-	-	0.03	
		α-terpinyl acetate	-	0.22	-	-	-	-	-	-
		myrtenyl acetate	-	0.03	-	-	-	-	-	-
Sesquiterpenoids	Acyclic Sesquiterpene	*trans*-nerolidol	-	0.42	0.73	11.46	-	6.46	1.21	0.35
		farnesol	-	-	-	-	-	1.17	-	-
	Monocyclic Sesquiterpene	β-bisabolene		-	-	-	-	-	0.24	-
	Bicyclic Sesquiterpene	β-caryophyllene	3.23	3.23	11.1	-	16.79	11.59	15.26	9.0
		caryophyllene oxide	-	-	-	-	2.4	0.65	1.07	2.09
		guaiol	4.48	-	-	8.19	-	-	-	-
		bulnesol	1.73	-	-	-	-	-	-	-
		δ-cadinene	-	13.63	2.01	-	-	-	-	-
		τ-cadinol	0.18	0.41	1.68	1.35	1.06	2.38	0.85	
		α-cadinol	1.0	0.76	-	-	1.13	4.94	0.29	
		γ-eudesmol	1.84	-	-	-	-	-	-	0.4
		α-Epi-7-epi-5-eudesmol	3.1	1.4	-	-	2.03	-	-	-
		globulol	-	-	-	1.06	-	-	-	-
		ledol	0.87	-	-	-	-	-	-	-
		β-copaene	-	4.21	3.12	-	1.18	-	0.74	-
		α-copaene	3.26	3.16	0.74	5.25	3.94	5.89	-	1.53
		β-elemene	-	0.78	-	-	-	-	-	-
		γ-elemene	0.96	-	-	0.13	0.57	-	0.11	-
		δ-elemene	0.99	0.12	-	-	-	-	-	-
		(−)-*cis*-β-elemene	9.08	-	2.30	9.63	6.58	-	23.59	1.81
		α-cubebene	0.74	1.73	-	14.3	-	1.21	-	-
		(−)-α-gurjunene	0.09	0.84	-	5.22	0.29	0.15	-	-
		γ-gurjunene	0.72	-	-	0.8	0.33		-	-
		alloaromadendrene	-	-	-	2.85	-	-	-	-
		isoaromadendrene epoxide	-	-	8.4	0.18	-	-	-	-
		humulene epoxide II	-	1.47	1.62	1.14	-	2.19	0.13	-
		spathulenol	-	-	-	-	-	-	-	-
		isospatulenol	-	-	-	-	-	3.89	-	-
		γ-muurolene	1.2	8.26	-	-	0.73	2.31	2.24	0.41
		α-muurolene	2.17	0.83	-	-	1.08	0.8	1.09	-
		δ-amorphene	3.77	0.37	-	-	14.58	4.44	10.15	2.11
		α-selinene	-	-	-	4.51	-	-	-	-
		β-selinene	0.41	-	-	-	-	-	1.88	-
		γ-selinene	-	-	-	1.60	-	-	2.43	-
		β-calacorene	-	0.71	-	1.1	0.45	0.38	0.17	-
		*trans*-calamenene	0.17	1.31	-	1.46	0.31	0.89	-	0.47
		cadalene	-	-	-	2.08	-	-	-	-
	Tricyclic Sesquiterpene	α-cedrene	-	-	-	-	-	0.21	-	-
		ylangene	-	0.55	-	-	-	-	-	-
		sesquiterpene alcohols (hedycaryol)	10.86	-	-	-	0.96	-	-	0.34
		(1a*R*,4a*R*,7S,7a*R*,7b*R*)-1,1,7-trimethyl-4-methylenedecahydro-1*H*-cyclopropa[e]azulen-7-ol	1.63	3.73	10.04	3.62	6.13	15.0	0.82	1.67
		(3*R*,3a*R*,3b*R*,4*S*,7*R*,7a*R*)-4-isopropyl-3,7-dimethyloctahydro-1*H*-cyclopenta[1,3]cyclopropa[1,2]benzen-3-ol	0.49	2.34	6.63	2.17	0.66	0.72	0.19	2.03
		7*R*,8*R*-8-hydroxy-4-isopropylidene-7-methylbicyclo[5.3.1]undec-1-ene	-	0.16	-	0.59	-	2.76	-	-
		eudesm-7(11)-en-4-ol	-	-	-	1.67	-	-	-	-
		selin-6-en-4α-ol	-	-	-	-	-	-	0.27	-
		neointermedeol	-	-	-	-	-	-	0.61	-
		(+)-isovalencenol	-	-	-	-	-	-	-	0.25
	Sesquiterpene Ketones	dehydrofukinone	-	-	-	-	0.35	-	-	6.82
		salvial-4(14)-en-1-one	-	-	-	-	-	0.85	-	-
		β-vatirenone	-	-	-	-	-	-	-	0.56
Diterpenoids	Acyclic Diterpene	phytol	-	-	-	0.19	3.54	0.97	-	-
		hexahydrofarnesyl acetone	-	-	-	0.06	-	-	-	-
		phytone	-	-	-	-	-	-	-	0.15
Phenylpropanoids	Allylbenzenes	myristicin	-	0.15	38.26	-	-	-	0.29	-
		apiol	-	7.21	3.08	-	2.19	0.49	14.08	1.01
Fatty Acids and Derivatives	Fatty Acids	palmitic acid	-	-	-	-	0.56	0.4	-	-
	Fatty Alcohols	1-octadecanol	-	-	-	0.09	-	-	-	-
	Fatty Aldehydes	pentadecanal	-	-	-	-	-	-	0.08	-
	Fatty Esters	methyl palmitate	-	-	-	-	0.13	-	-	-
		homosalate	-	-	-	-	0.19	-	-	-
Benzenoids	Phenols	phenol, 2,4-bis(1,1-dimethylethyl)-6-methyl-	-	-	-	-	-	-	-	10.01
		thymol	-	-	-	-	-	-	0.08	-
	Naphthalenes	1,2,9,10-tetradehydroaristolane	-	-	-	0.97	-	-	-	-
		agarospirole	-	-	-	-	-	-	-	2.14
		α-agarofuran	-	-	-	-	-	-	-	0.23
		dihydroagarofuran	-	-	-	-	-	-	-	8.08
		2-tert-butylquinoline	-	-	-	-	1.01	-	-	-
Other Compounds	Alkanes	pentadecane	-	-	5.86	-	-	-	-	-
		1,4-diisopropylbenzene	-	-	-	-	0.73	-	-	-
	Alkenes	2-methyl-1-pentene	-	-	-	-	-	-	16.7	-
		1,4-dimethyl-4-vinylcyclohexene	-	-	-	-	-	0.15	-	-
	Ketones	6-methyl-5-hepten-2-one	-	-	-	0.49	-	-	-	-
		6-methyl-3,5-heptadien-2-one	-	-	-	0.01	-	-	-	-
	Aldehydes	benzaldehyde	-	-	-	0.01	-	-	-	-
	Alcohols	*cis*-3-hexen-1-ol	-	-	-	-	-	0.12	-	-
		linalool	-	-	-	-	0.14	0.13	0.12	-
	Esters	3,8-dimethyl-5-α-hydroxy-δ^9-octa-hydroazulene acetate	-	0.93	-	-	-	-	-	8.74
		5-azulenemethanol	0.23	-	-	-	0.67	-	-	-
		(1*S*,3*S*,5*S*)-1-isopropyl-4-methylenebicyclo[3.1.0]hexan-3-yl acetate	-	0.02	-	-	-	-	-	-
	Ethers	liguloxide					1.16			
	Miscellaneous	oplopenone	-	-	-	-	-	-	-	0.97
		oxo-tremorine	-	-	-	-	0.64	-	0.21	-
		teaspirane	-	0.06	-	-	-	-	-	-
		peruviol	0.07	-	-	-	6.23	-	-	-
		2-propenoic acid, 3-[4-[(3-methyl-1-butenyl)oxy]phenyl]-, methyl ester	-	-	-	-	1.95	-	-	-

Note: Compound name identified by GC/MS compared to NIST library 17. Relative abundance (%): relative amount of identified compounds as a function of the area of each peak in the total area of the chromatogram (Appendix A).

**Table 2 molecules-30-04236-t002:** In vitro antibacterial activity of essential oils by disk diffusion method.

Essential Oils	Microorganism											
	*E. coli*				*S. enteritidis*				*S. aureus*			
	I.C.A. (mm)	I.E.O. (mm)	RI (%)	Act	I.C.A. (mm)	I.E.O. (mm)	RI (%)	Act	I.C.A. (mm)	I.E.O. (mm)	RI (%)	Act
*E. cornuta*	19.43 ± 4.93	6.48 ± 0.47	33.51	L	24.00 ± 8.19	10.92 ± 4.98	45.51	L	19.33 ± 2.89	6.33 ± 0.58	32.76	L
*M. jaenensis*	17.00 ± 2.65	7.11 ± 1.06	41.83	L	16.67 ± 0.58	6.56 ± 0.96	39.33	L	35.00 ± 0.00	35.00 ± 0.00	100	H
*M. manguillo*	14.67 ± 1.53	6.24 ± 0.37	42.58	L	15.00 ± 6.08	7.33 ± 2.03	48.89	L	10.67 ± 1.15	6.49 ± 0.73	60.83	I
*P. aduncum*	22.00 ± 4.36	6.11 ± 0.19	27.78	L	18.67 ± 2.52	6.33 ± 0.58	33.93	L	23.00 ± 7.55	6.00 ± 0.00	26.09	L
*P. amalago*	20.00 ± 3.61	7.04 ± 3.02	35.22	L	20.33 ± 0.58	6.60 ± 0.93	32.46	L	33.33 ± 2.89	33.56 ± 2.50	100.67	H
*P. glabribaccum*	19.67 ± 9.61	6.06 ± 0.10	30.79	L	20.00 ± 3.46	6.92 ± 1.51	34.61	L	16.00 ± 16.5	15.89 ± 16.6	99.31	H
*T. integrifolia*	14.00 ± 1.00	6.40 ± 0.32	45.71	L	17.00 ± 5.57	7.47 ± 1.62	43.92	L	35.00 ± 0.00	35.00 ± 0.00	100	H
*Z. fagara*	20.00 ± 4.58	6.46 ± 0.34	32.28	L	20.33 ± 9.07	6.89 ± 1.54	33.88	L	16.00 ± 0.00	10.04 ± 3.98	62.78	I

I.C.A.: Inhibition (mm) of control antibiotic (Streptomycin); I.E.O.: Inhibition (mm) of essential oil, RI (%): Relative inhibition percentage (%). Activity: (L) low, (I) intermediate, and (H) high. Mean ± standard deviation.

**Table 3 molecules-30-04236-t003:** Antioxidant capacity and total content of phenolic compounds of the essential oils.

Essential Oils	Antioxidant Activity			TPC (mg GAE/g)
	DPPH (µmol TE/g)	FRAP (µmol Fe^2+^/g)	ABTS (µmol TE/g)	
*E. cornuta*	7.72 ± 0.08 e	39.39 ± 2.03 f	9.91 ± 0.19 e	67.74 ± 1.08 g
*M. jaenensis*	8.61 ± 0.21 d	73.69 ± 2.80 c	16.43 ± 0.13 b	96.09 ± 1.65 d
*M. manguillo*	7.88 ± 0.14 e	71.19 ± 1.97 c	8.44 ± 0.10 f	113.65 ± 0.13 c
*P. aduncum*	19.28 ± 0.09 a	111.79 ± 0.78 a	19.02 ± 0.09 a	132.64 ± 0.23 b
*P. amalago*	7.61 ± 0.08 e	22.59 ± 1.55 h	8.86 ± 0.16 f	50.11 ± 0.77 h
*P. glabribaccum*	7.20 ± 0.07 f	64.99 ± 0.81 d	7.86 ± 0.10 g	86.81 ± 1.68 e
*T. integrifolia*	11.12 ± 0.10 c	104.64 ± 1.29 b	15.20 ± 0.10 c	159.34 ± 0.19 a
*Z. fagara*	7.16 ± 0.09 e	32.86 ± 1.15 g	7.86 ± 0.41 g	52.49 ± 0.06 h

Data are presented as mean ± standard error. Different letters represent significant differences between treatments, according to Tukey’s test (*p* < 0.05).

**Table 4 molecules-30-04236-t004:** Indicators obtained by principal component analysis (PCA) for extraction yield, antioxidant activity, and antimicrobial activity of the essential oils of the eight plant species.

Essential Oils	Extraction Yield Cumulative Var: 94%	Antioxidant Activity Cumulative Var: 97%	Antimicrobial Activity Cumulative Var: 94%
*E. cornuta*	20.18	78.07	34.49
*M. jaenensis*	8.82	122.10	102.39
*M. manguillo*	14.71	133.42	63.21
*P. aduncum*	79.64	175.14	27.65
*P. amalago*	95.64	53.79	102.61
*P. glabribaccum*	58.27	108.82	100.73
*T. integrifolia*	22.28	190.35	102.64
*Z. fagara*	176.44	62.06	64.59

Contributions of observations (%) of EOs to the first principal components of PCA. The analysis was performed on an integrated matrix containing all variables (yield, antioxidant, and antibacterial activity). The cumulative variance of the first component for each set of variables is indicated in the header.

## Data Availability

Dataset available on request from the authors.

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
