# Peer review of "Discovering the Bioactive and Antibacterial Potential of Essential Oils from Aromatic Plants of Northeastern Peru"

_molecules, 2025, doi:10.3390/molecules30214236_

Round 1
Reviewer 1 Report
Comments and Suggestions for Authors
The manuscript entitled "Seasonal variation of extraction yield, in vitro biological activity and chemical composition of essential oils of plant species from northeastern Peru" deals with chemical analysis and in vitro biological (antimicrobial and antioxidant) activity of eight essential oils isolated by hydrodistillation from the leaves of certain members of the Peruvian spontaneous flora.
The presentation of the results is generally good, with some omissions that can be easily corrected. But, there are some methodological concerns that prevent me from recommending it for publication in its current form.
First, the Introduction section provides some generalized statements about the facts that are already well-known to the readership, but fails to address the main issue: what were the selection criteria for including selected plant species in this investigation? Was the selection based on ethnopharmacological importance or something else? The study begins with 29 plants, but essential oils were obtained from only 10. Finally, only 8 were selected for full analysis, but the criteria for this stepwise reduction are not clear. I guess that it would be better to devote the whole study for those eight essential oils only.
Second, the use of undiluted essential oils without a proper solvent control is at least unusual practice and makes it difficult to interpret and compare the results of the antimicrobial activity investigations. In my opinion, the authors should explain where appropriate (Results and Discussion) why this technique was applied.
Finally, the text at the very beginning of section 3. Materials and Methods, page 12, lines 306-312, should be deleted.
Author Response
The manuscript entitled "Seasonal variation of extraction yield, in vitro biological activity and chemical composition of essential oils of plant species from northeastern Peru" deals with chemical analysis and in vitro biological (antimicrobial and antioxidant) activity of eight essential oils isolated by hydrodistillation from the leaves of certain members of the Peruvian spontaneous flora. The presentation of the results is generally good, with some omissions that can be easily corrected. But, there are some methodological concerns that prevent me from recommending it for publication in its current form.
Response: Thank you for your comments. We have made the changes according to your feedback (changes in red).
First, the Introduction section provides some generalized statements about the facts that are already well-known to the readership, but fails to address the main issue:
what were the selection criteria for including selected plant species in this investigation? Was the selection based on ethnopharmacological importance or something else?
Response: Thank for the suggestion. We added the selection criterion [L. 94-101]
- The study begins with 29 plants, but essential oils were obtained from only 10. Finally, only 8 were selected for full analysis, but the criteria for this stepwise reduction are not clear. I guess that it would be better to devote the whole study for those eight essential oils only.
Response: Sorry for the mistake. We updated the information in the main text [L. 27; 102-103; 348-354]
Second, the use of undiluted essential oils without a proper solvent control is at least unusual practice and makes it difficult to interpret and compare the results of the antimicrobial activity investigations. In my opinion, the authors should explain where appropriate (Results and Discussion) why this technique was applied.
Response: Thank for you comment. We added information explaining the use of the technique in the study [L. 243-255]
Finally, the text at the very beginning of section 3. Materials and Methods, page 12, lines 306-312, should be deleted.
Response: Thank for the suggestion. We deleted the sentence [L. 346-347]
Reviewer 2 Report
Comments and Suggestions for Authors
Dear colleagues, I believe your study addresses an interesting topic; however, there are some aspects that need to be improveb before it can be considered for publication. Please find my detailed comments and suggestions in the attached PDF file. Some of my main concerns are:
- The continuous reference to 29 plant species throughout the manuscript is misleading, as the study was actually conducted with only 8 species (and not even the 10 mentioned elsewhere).
- Some sections lack clarity, some sentences are too long. Puntuaction need correction in some sections.
- Statistical analyses (ANOVA and Tukey), data from Table 3.
- Discussion should be improved. I think you can benefit more from your results.

The English language is generally fine, but several sections lack clarity, and some sentences are too long and difficult to understand. I recommend a thorough revision.
Author Response
Dear colleagues, I believe your study addresses an interesting topic; however, there are some aspects that need to be improveb before it can be considered for publication. Please find my detailed comments and suggestions in the attached PDF file.
Response: Thank you for your comments. We have made the changes according to your feedback (changes in red).
Some of my main concerns are:
The continuous reference to 29 plant species throughout the manuscript is misleading, as the study was actually conducted with only 8 species (and not even the 10 mentioned elsewhere).
Response: Sorry for the mistake. We updated the information in the main text [L. 27; 102-103; 348-354]
Some sections lack clarity; some sentences are too long. Punctuation need correction in some sections.
Response: Thank for you comment. We corrected the sentences for clarity and punctuation in some sections [L. 284-298; 313-320]
Statistical analyses (ANOVA and Tukey), data from Table 3.
Response: Thank for the suggestion. We added the required information [L. 272-281]
Discussion should be improved. I think you can benefit more from your results.
Response: Thank for the suggestion. We improved the discussion [L. 257-278; 294-311]
Comments on the attached file:
We have made all the wording and style changes according to your recommendations. The changes are highlighted in red. The main changes made are described below:
- As a suggestion, I would make the title more attractive.
Response: We improved the title [L. 2-3]
- I believe this is incorrect since EOs were extracted from only 8 species. Please focus your study on those 8 species. Remove all mention of the 29 species from abstract, introduction, materials and methods and results and discussion sections. It´s confusing to the reader and I don´t understand why reporting that you sampled 29 species if all the investigation was conducted with only 8.
Response: We updated the information [L. 27; 102-103; 348-354]
- Piper sp. belongs to the family Piperaceae. In the previous sentences, you discussed the importance of the family Asteraceae and mentioned the species Tesaria integrifolia. Then, in the following sentence, you introduced a plant genus from another family. With this structure, the reader might mistakenly interpret that Piper belongs to Asteraceae. Please improve the introduction section.
Response: We improved the section [L. 56-63]
- I would rewrite this sentence, for example "The first one implies the use of synthetic additives in intensive food production systems, given their low cost and immediate availability on the market"
Response: We improved the sentence [L. 76-78]
- For example, "The second trend is more recent and depends on the use of green technologies in food production systems to yield more natural products". These are marely suggestions to make these sentences more diirect and concise.
Response: We improved the section [L. 78-79]
- I believe that a paragraph with a definition of EOs and the background on their use as food additives given their advantages should be included. These advantages should consider the bioactivity against microorganisms, antioxidant activity and composition-phenolic content etc.
Response: We improved the section [L. 84-93]
- through the disk difussion assay
Response: We improved the sentence [L. 107-108]
- It´s now that I understand that 29 plant species were collected, but essential oils were only extracted from 10, and 8 were tested. It is confusing for the reader to mention all 29 species in the abstract, in the last paragraph of the introduction, and the materials and methods section. Perhaps focusing only on the 8 species that were actually tested would make the presentation clearer and more concise. Even in the introduction it is confusing why you gave details of only 8 species and in the last paraghraph you mention all the 29 species. Besides, you give the reference to Table 1, and that table only report the final 8 species, not 10.
Response: We updated the information [L. 27; 102-103; 348-354]
- You mention that caryophyllene and derivative are predominant but that information is not given in Table 1. I suggest replacing "X" by relative percentages. Also, which other compounds is included in "caryophyllene derivatives" besides caryophyllene oxide? Because you say it in plural.
Response: We updated the information [L.153; 160-165]
- scientific names in italics please. Check throughout the manuscript.
Response: We updated the information [L. 156; 170; 133; 135; 223-225; 259]
- Table 1: I understand that authors have included relative percentages of compounds from each EO in Supplementary Material. I strongly suggest including relative percentages in Table 1. It is not clear if the compounds listed here are those considered as main compounds (> 5%). Please include this information in the Table footnote.
Response: We improved the Table 1 [L. 160-165]
- I believe you are still discussing P. aduncum? SHould be in the same paragraph not a new one.
Response: We improved the sentence [L. 178-183]
- Even though the chemical composition of P. aduncum is well compared with existing literature, I believe that the link between the chemical composition and geographical origin of the plants (environmental parameters, growth seasons, part of the plants from where the EOs were extracted) is missing. This would be rather important, particularly in the context of chemotaxonomy stated by the authors. "This organized data is crucial for identifying chemotaxonomic patterns and potential biological activities linked to specific compound classes across these native Peruvian plants." Please include these aspects for all the species tested (discussed in the following paragraphs)
Response: We improved the section [L. 166-217]
- In Materials and Methods Section you first describe the methodology on extraction and yield of EOs but in the results and discussion section you first adress the chemical composition. Please change.
Response: We modified the section [L. 113-151]
- I suggest including in this section literature on the antibacterial activity of pure compounds corresponding to the major compounds of the EOs with highest antibacterial activity.
Response: We improved the section [L. 230-255]
- Please improve the writing of this section (2.5. Grouping of the best performing essential oils in terms of antibacterial activity and antioxidant activity).
Response: We improved the section [L. 313-345]
- The meaning of 'majority of compounds' is not entirely clear. In chemical terminology, major compounds refers to those with the highest relative percentages, i.e., the most abundant (also called main compounds), whereas 'majority of compounds' could be understood as the number of compounds shared (or not) among species. The distinction should be clarified.
Response: We clarified the section [L. 313-335]
- This is an interesting line of reasoning to explore. Figure 3 illustrates the differentiation of EOs in terms of chemical composition. Considering the distribution of compounds in the biplot together with the biological activities of EOs from the previous analysis, patterns may emerge. For example, Mj and Pg showed good antimicrobial activity and had copaene and nerolidol as their main constituents. It would be interesting to search the literature for evidence supporting such causality (e.g., nerolidol, copaene, or EOs with these compounds as major constituents exhibiting antimicrobial activity). This kind of reasoning would help to integrate the results.
Response: We improved the section [L. 313-320]
- Is this a PCA? Were all the compounds considered in the analysis? or only the main compounds of EOs (>5%)?
Response: We clarified the section [L. 313-315]
- I only counted 27 in Figure 4. Was the same species collected in two different points? If so please detail.
Response: We clarified the section [L. 348-354]
Reviewer 3 Report
Comments and Suggestions for Authors
Summary
The manuscript entitled “Seasonal variation of extraction yield, in vitro biological activity and chemical composition of essential oils of plant species from northeastern Peru” explores the extraction, chemical characterization, and biological activities of essential oils (EOs) from plants collected in Cajamarca, Peru. The topic is relevant to natural products research and could contribute to the understanding of native/endemic plant resources with potential applications in food and pharmaceutical industries.
However, the manuscript in its current form suffers from incomplete reporting of chemical composition, inconsistencies in experimental scope versus results, and missing supporting data that are essential for reproducibility and validation.
Strengths
- The study focuses on largely unexplored native and endemic species of northeastern Peru, some reported for the first time.
- Seasonal variation in yield is addressed, which adds ecological and applied value.
- Antioxidant and antibacterial activities were systematically evaluated with statistical comparisons.
- The attempt to integrate PCA and clustering to classify oils based on activity and composition is commendable.
Major Concerns
- Mismatch between scope and presented results
- The abstract and methods indicate that EOs were obtained from 29 species, yet only 8 species are presented in the results for chemical composition, antioxidant, and antibacterial activities.
- The rationale for excluding the other 21 species is not explained. Either results for all species must be included, or the abstract and objectives must be revised to accurately reflect the study’s scope.
- Incomplete chemical characterization
- Although n-alkane standards were mentioned, Linear Retention Index (LRI) values (experimental and literature) are not presented.
- Match factors (%) from MS library searches (e.g., NIST) are missing. Without this information, compound identification remains insufficiently validated.
- Only Table S1 is referenced in SI, but chromatograms and spectra are absent.
- Missing chromatographic and spectral data
GC–MS chromatograms for each EO and mass spectra of identified compounds must be included in the supplementary materials. This is standard in essential oil characterization studies and necessary for reproducibility.
- Antibacterial activity methodology
- Only the disc diffusion assay was used. This provides limited information and does not quantify antibacterial potency.
- No MIC or MBC values are reported. If not feasible, this limitation must be explicitly acknowledged, and claims about antibacterial efficacy should be moderated.
- PCA and clustering methods
- The PCA description is unclear: Was it performed separately for yield, antioxidant, and antibacterial activity, or combined?
- The justification for including only 8 oils in PCA, despite extraction from 29, is missing.
- The meaning of the “cumulative variance” values in Table 4 is not well explained.
Minor Concerns
- Nomenclature and consistency
- Plant species names are inconsistently written (Piper amalogo vs. P. amalago; glabribaccum vs. glabribacum). These should be standardized throughout.
- Abbreviations should be defined at first use (e.g., EO, PCA, PIR).
- Statistical analysis
- Table 3 uses superscript letters for Tukey’s test results, but the explanation is confusing. Clarify the exact sample size (n) and confirm that ANOVA assumptions were met.
- All figure legends should clearly define abbreviations and variables.
- Materials and Methods clarity
- Extraction yield description lacks information on replication (number of independent extractions per species).
- Antioxidant and antibacterial tests mention triplicates in some cases but not consistently across all assays. Please clarify.
Recommendation
I recommend major revisions before this manuscript can be considered for publication. The study has potential scientific value, but substantial additions and clarifications are necessary, particularly in terms of chemical characterization, completeness of reported data, and methodological transparency.
Author Response
The manuscript entitled “Seasonal variation of extraction yield, in vitro biological activity and chemical composition of essential oils of plant species from northeastern Peru” explores the extraction, chemical characterization, and biological activities of essential oils (EOs) from plants collected in Cajamarca, Peru. The topic is relevant to natural products research and could contribute to the understanding of native/endemic plant resources with potential applications in food and pharmaceutical industries. However, the manuscript in its current form suffers from incomplete reporting of chemical composition, inconsistencies in experimental scope versus results, and missing supporting data that are essential for reproducibility and validation.
Response: Thank you for your comments. We have made the changes according to your feedback (changes in red).
Strengths: The study focuses on largely unexplored native and endemic species of northeastern Peru, some reported for the first time. Seasonal variation in yield is addressed, which adds ecological and applied value. Antioxidant and antibacterial activities were systematically evaluated with statistical comparisons. The attempt to integrate PCA and clustering to classify oils based on activity and composition is commendable.
Response: Thank for you comment.
Major Concerns
- Mismatch between scope and presented results
Response: Thank for you comment. We have improved the manuscript to clarify the scope and the results presented [L. 113-345]
- The abstract and methods indicate that EOs were obtained from 29 species, yet only 8 species are presented in the results for chemical composition, antioxidant, and antibacterial activities.
Response: Sorry for the mistake. We updated the information in the main text [L. 27; 102-103; 348-354]
- The rationale for excluding the other 21 species is not explained. Either results for all species must be included, or the abstract and objectives must be revised to accurately reflect the study’s scope. Incomplete chemical characterization
Response: Sorry for the mistake. We updated the information in the main text [L. 27; 102-103; 348-354]
- Although n-alkane standards were mentioned, Linear Retention Index (LRI) values (experimental and literature) are not presented.
Response: Thank for you comment. We added present the values of the linear retention index (LRI) [L. 476-478]
- Match factors (%) from MS library searches (e.g., NIST) are missing. Without this information, compound identification remains insufficiently validated.
Response: Thank for the suggestion. We added the match factors (%) from MS library searches (e.g., NIST) are missing [L. 476-478]
- Only Table S1 is referenced in SI, but chromatograms and spectra are absent. Missing chromatographic and spectral data
Response: We added the chromatograms and spectra [L. 476-478], and Table S2
- GC–MS chromatograms for each EO and mass spectra of identified compounds must be included in the supplementary materials. This is standard in essential oil characterization studies and necessary for reproducibility.
Response: We added the chromatograms and spectra [L. 476-478], and Table S2
Antibacterial activity methodology
- Only the disc diffusion assay was used. This provides limited information and does not quantify antibacterial potency. No MIC or MBC values are reported. If not feasible, this limitation must be explicitly acknowledged, and claims about antibacterial efficacy should be moderated.
Response: Thank you for your comment. We added the relevant information about this section and suggested new studies as a recommendation, complementing them with the techniques indicated [L. 243-255]
PCA and clustering methods
- The PCA description is unclear: Was it performed separately for yield, antioxidant, and antibacterial activity, or combined?
Response: Thank. We improved the PCA description according to comments [L. 282-311]
- The justification for including only 8 oils in PCA, despite extraction from 29, is missing.
Response: Sorry for the mistake. We updated the information in the main text [L. 282-311]
- The meaning of the “cumulative variance” values in Table 4 is not well explained.
Response: Thank for the suggestion. We have improved the explanation [L. 288-290; 301-303]
Minor Concerns:
Nomenclature and consistency
- Plant species names are inconsistently written (Piper amalogo vs. P. amalago; glabribaccum vs. glabribacum). These should be standardized throughout.
Response: Sorry for the mistake. We standardized the names [L. 36;41
- Abbreviations should be defined at first use (e.g., EO, PCA, PIR).
Response: We defined the terms [L. 225; 285]
Statistical analysis
- Table 3 uses superscript letters for Tukey’s test results, but the explanation is confusing. Clarify the exact sample size (n) and confirm that ANOVA assumptions were met.
Response: We clarified the exact sample size (n) and confirm that ANOVA assumptions were met [L. 269-278]
- All figure legends should clearly define abbreviations and variables
Response: We defined the abbreviations and variables [L. 307-311; 324-335]
Materials and Methods clarity
- Extraction yield description lacks information on replication (number of independent extractions per species).
Response: We added the information on replication (number of independent extractions per species) [L. 114-116]
- Antioxidant and antibacterial tests mention triplicates in some cases but not consistently across all assays. Please clarify.
Response: We have added the required information [L. 402-403; 427-428; 438; 448]
Recommendation: I recommend major revisions before this manuscript can be considered for publication. The study has potential scientific value, but substantial additions and clarifications are necessary, particularly in terms of chemical characterization, completeness of reported data, and methodological transparency.
Response: Thank you for your comments. We have made the changes according to your feedback (changes in red).
Round 2
Reviewer 2 Report
Comments and Suggestions for Authors
Authors have significantly improved their manusript according to the suggested changes. However, there are minor issues that should be addressed before publication.
1- Table S2: names of pure compounds should be in English.
2- Improve the quality of figures, particularly the PCAs.
3- Regarding English language: results should be reported in Past Tense, only when you refer to a Table or a Figure you use Present. Other corrections were made on the manuscript.
4- Table 1. Check that some values have a single decimal, not two. Also, the names of the table headers should not be truncated.
5- Other comments were made in the pdf file.

In general, the English language is fine. However, I recommend a revision by a native speaker to improve the academic writing. You can also check the Academic Phrasebank.
Author Response
Authors have significantly improved their manuscript according to the suggested changes. However, there are minor issues that should be addressed before publication.
Response: Thank you for your comments. We have addressed your suggestions in this second round of manuscript improvements (changes in red).
- Table S2: names of pure compounds should be in English.
Response: Sorry for the mistake. We wrote the names in English (see to Table S2)
- Improve the quality of figures, particularly the PCAs.
Response: Thank for the suggestion. We improved the figures [L. 119-120, 287-289, 305-307, 345-346, 351-352]
- Regarding English language: results should be reported in Past Tense, only when you refer to a Table or a Figure you use Present. Other corrections were made on the manuscript.
Response: Thank for the comments. We corrected the language (changes in red).
- Table 1. Check that some values have a single decimal, not two. Also, the names of the table headers should not be truncated.
Response: Thank, we updated the table according the suggestion [L. 150-154]
- Other comments were made in the pdf file.
Response: Thank for the suggestions. We modified in the main text (changes in red)